# Influence of Fermented Red Clover Extract on Skeletal Muscle in Early Postmenopausal Women: A Double-Blinded Cross-Over Study

**DOI:** 10.3390/nu12113587

**Published:** 2020-11-23

**Authors:** Mikkel Oxfeldt, Line Barner Dalgaard, Jeyanthini Risikesan, Frank Ted Johansen, Mette Hansen

**Affiliations:** 1Department of Public Health, Aarhus University, 8000 Aarhus C, Denmark; mox@ph.au.dk (M.O.); lbdalgaard@ph.au.dk (L.B.D.); fjpoohcorner@hotmail.com (F.T.J.); 2Department of Clinical Medicine, Diabetes and Hormones Diseases, Aarhus University Hospital, 8200 Aarhus N, Denmark; jeyanrk@clin.au.dk

**Keywords:** phytoestrogen, dietary supplement, isoflavones, muscle protein signaling, anabolic signaling

## Abstract

**Objective:** To investigate effects of supplementation with a fermented red clover (RC) extract on signaling proteins related to muscle protein synthesis and breakdown at rest and in response to a resistance exercise bout. **Methods:** Ten postmenopausal women completed a double-blinded cross-over trial with two different intervention periods performed in random order: (A) RC extract twice daily for 14 days, and (B) placebo drink twice daily for 14 days. The intervention periods were separated by a two-week washout period. After each intervention period a muscle tissue sample was obtained before and three hours after a one-legged resistance exercise bout. Muscle strength was assessed before and after each intervention period. **Results:** Protein expression of FOXO1 and FOXO3a, two key transcription factors involved in protein degradation, were significantly lower and HSP27, a protein involved in cell protection and prevention of protein aggregation was significantly higher following RC extract compared to placebo. No significant treatment × time interaction was observed for muscle protein expression in response to exercise. However, p-mTOR, p-p70S6k and HSP90 protein content were significantly increased in response to exercise in both groups. **Conclusion:** This study demonstrates that RC extract supplementation downregulates molecular markers of muscle protein degradation compared to placebo in postmenopausal women.

## 1. Introduction

Sarcopenia, defined as an age-associated decline in skeletal muscle mass, correlates with higher rates of falls, bone fractures and mortality leading to high health care costs [1], and reduced quality of life [2].

During the transition into menopause, women experience an accelerated decline in muscle mass and strength [3], which has been coupled to the accompanying marked reduction in estrogen. The reduction in muscle mass and strength is critical as women in general have a lower muscle mass and a longer life expectancy compared to men [4], making them particularly vulnerable to becoming frail as age increases. Hence, counteracting the decline in muscle mass and strength in this population seems vital.

Progressive resistance training is an effective strategy for increasing muscle mass and strength in the elder population [5]. However, the effectiveness of training seems to be lowered in postmenopausal women, possibly related to the low circulating concentration of estrogen [6,7,8]. Use of hormone therapy (HT) has been shown to have beneficial effects on skeletal muscle [7], by increasing muscle strength [9] and enhancing the response to resistance training [10]. Nevertheless, HT is also known to be associated with an increased risk of cancer in estrogen receptor (ER)-α rich tissues (i.e., breast, ovaries and endometria) [11,12,13,14]. A recent meta-analysis reported a pooled hazard ratio/relative risk for ovarian cancer and HT use to be 1.55 (95% CI: 1.05 to 2.30) [14]. Consequently, there is an urgent need to develop safe and effective alternative therapies to HT.

Isoflavones are the most common and most potent type of phytoestrogen; polyphenolic non-steroidal compounds with estrogenic activity [15]. They are found in legumes, such as soy and red clover (RC) [16] and mimic the actions of endogenous estrogen through ER binding and activation of ER-dependent gene transcription [15]. However, compared to estrogen, which has relative high binding affinity for the ER-α and ER-β, isoflavones demonstrate up to ~1600 times lower affinity for ER-α [17]. Thus, isoflavones are promising candidates for substituting traditional HT without increasing the risk of cancer in ER-α rich tissues. Accordingly, epidemiological studies show no association between intake of isoflavones and estrogen related cancer [18,19] and a recent study performed in a large multiethnic cohort (*n* = 84.450) indicated that higher isoflavone intakes may protect against estrogen related cancer in Latina, African American and Japanese American women [20].

The structural integrity of isoflavones, allow them to selectively activate the ER-β and therefore, mainly induce positive estrogenic effects in ER-β rich tissues [17]. ER-β is expressed and localized within skeletal muscle tissue [21], and ER-β activation appears to regulate signaling pathways important for skeletal muscle growth and regeneration [22]. In animals, a number of studies have investigated the influence of isoflavones found in soy (mainly genistein and daidzein) on skeletal muscle [23,24,25,26]. The results are promising, showing reduced muscle atrophy [23,24], lower expression of ubiquitin-specific protease 19 through ER-β activation [25], and an improved anabolic response to training in rats, accompanied by higher expression of myogenic regulatory factors [26]. In a controlled human trial, six months of soy isoflavone supplementation increased lean body mass in obese-sarcopenic postmenopausal women [27]. However, studies investigating a possible additive effect of isoflavones on training adaptations have found no effect on lean body mass [28,29,30,31] or muscle strength [30,31], but reduced fat mass [28,29]. Interestingly, while most studies have focused on soy isoflavones, little attention has been given to isoflavones derived from RC. Compared to soy, RC contains high amounts of formononetin and biochanin A [17], and red clover derived products have shown to elicit higher estrogenic activity compared to those made from soy [32]. The fermentation of RC by probiotic lactic acid bacteria converts isoflavone glycosides to aglycones, which vastly enhances bioavailability [33]. So far, only two randomized controlled trials [34,35] have examined the effects of fermented RC, demonstrating that daily intake of RC extract improves bone status in postmenopausal women following three [35] and 12 months [34]. There are currently no reports of clinical trials investigating the effects of RC extract on skeletal muscle. Collectively, the unique formula of fermented RC extract makes it a promising candidate for an alternative to HT. Therefore, the aim of the present pilot study was to investigate the effect of 14 days of fermented RC extract supplementation on signaling proteins related to muscle protein synthesis and breakdown at rest and in response to a resistance exercise bout. A secondary aim was to elucidate whether RC extract supplementation would affect muscle strength. We hypothesized that RC extract supplementation would positively influence skeletal muscle by increasing muscle strength and the anabolic response to a resistance exercise bout.

## 2. Methods

### 2.1. Ethical Approval

The present study was carried out at the Department of Public Health, Aarhus University, Denmark, and was in accordance with the Declaration of Helsinki, approved by the Central Denmark Region Committees on Health Research Ethics (1-10-72-212-19) and registered at Clinical.trials.gov (ID: NCT04154206). All participants signed an informed consent before their enrolment into the study.

### 2.2. Design

In a double-blinded cross-over trial subjects (*n* = 10) completed two different intervention periods in random order: (A) Intake of fermented RC extract twice daily for 14 days, and (B) intake of a placebo drink (PLA) twice daily for 14 days. The intervention periods were separated by a two-week washout period (Figure 1). The randomization was based on recruitment order and participants and test personnel were unaware of the treatment allocation.

On day 1 and day 14 of each 14-day intervention period, subjects met in the laboratory to perform a maximal handgrip and maximal elbow flexor strength test. Additionally, on day 14 a muscle tissue sample was obtained before and three hours after a resistance exercise bout performed on one-leg. In the second intervention period, exercise and biopsies were performed in the contralateral leg. Prior to the exercise bout, subjects received the RC or PLA drink and immediately after exercise they consumed 25 g of whey protein. All tests were performed at the same time of the day and by the same test personnel at all test days. To standardize activity and diet prior to the experimental days, subjects were instructed to (1) abstain from any strenuous and/or unaccustomed activity 48 h before the experimental days, (2) monitor their number of steps using a step counter before experimental day 14A and replicate the number of steps on the day before experimental day 14B, (3) eat a similar meal the night before each experimental day and (4) meet in the laboratory after an overnight fast.

### 2.3. Participants

Healthy untrained early postmenopausal women were recruited through local businesses and online posters on social media platforms (Table 1). Based on previous studies in postmenopausal women demonstrating the ERs and estrogen sensitivity to be altered with increasing time since menopause [36], we decided to include women, who were in the early postmenopausal state (no more than five years since last bleeding period).

Inclusion criteria were: (1) Healthy women >40 years of age with (2) a body mass index (BMI) <30 and (3) unaccustomed to resistance training (resistance trained less than once a month for the last six months). (4) Had not menstruated for the last six months, but, (5) no more than five years since last bleeding period. Exclusion criteria were: (1) Participated regularly in more than three hours of training/week, with exception of bike transport (<70 km per week), (2) had any diseases, injuries or used any medicine affecting skeletal muscle or performing the exercise session, (3) current user of HT or any form of isoflavone supplement.

Subjects, who met the criteria for participation (Figure 2) were invited to an introductory meeting which provided information about the risks and benefits related to participation in the study. Furthermore, the subjects completed a physical activity questionnaire, a dual-energy X-ray absorption scan to determine body composition and a familiarization session to the laboratory tests.

### 2.4. Experimental Drinks

The RC extract is a commercially available product produced by Herrens Mark Aps, consisting of juice from pressed RC plants mixed with probiotic lactic acid bacteria to facilitate cold fermentation. The process converts isoflavone glucosides to 90% aglycones, which improves bioavailability. Stevia and a natural sugar-free raspberry flavoring were added to mask flavor and appearance of the product. The placebo (PLA) drink consisted of water with added food coloring (ammoniated caramel, (Kavli)), and the same sweeteners to make it taste and appear like the RC product.

The RC extract and PLA drink were sealed and packed in identical boxes and coded with A or B. All participants and the research team were blinded to the content of the boxes throughout the study. Upon completion, a third party from Herrens Mark Aps informed the research team about the content of product A and B. This setup has previously been used successfully in another study [34].

The RC extract and PLA drink were handed out on day 1 of each intervention period. During the intervention period, participants were instructed to consume a daily dose of 120 mL, distributed to 60 mL in the morning and 60 mL in the evening. For the RC product, this was equivalent to a daily dose of minimum 60 mg isoflavones, (90% aglycones). At the end of each 14-day intervention period, participants returned the packing of their product to the research team to monitor compliance. Compliance was 100%.

### 2.5. Muscle Biopsy

On the last day (day 14) of each experimental period (A and B), the subjects arrived at the laboratory after overnight fasting and a resting muscle biopsy was obtained from the middle of the vastus lateralis, approximately two-thirds from the iliac crest to the patellar plateau using a Bergström needle with suction under local anesthesia. Using randomization (based on the recruitment order), subjects had the biopsy taken from either their dominant or non-dominant leg on day 14A and from the opposite leg on day 14B. The second biopsy collected three hours after a resistance exercise bout was sampled from a new incision hole 3 cm proximal to the first incision hole made for the resting biopsy. All muscle samples were dissected free of fat and connective tissue and immediately frozen in liquid nitrogen and stored at −80 °C until further analysis.

### 2.6. Resistance Exercise

Immediately after the first muscle biopsy was obtained, the subjects performed a five minute warm-up on a Monark 928E (Monark, Varberg, Sweden), before they were seated on a knee-extension machine (Technogym SpA, Gambettola, Italy) to begin the resistance exercise. Using the biopsy-leg only, subjects completed five sets of 10 repetitions at 10 repetition maximum (RM) with two minutes of rest between sets. The last set was performed to voluntary muscle failure. In case failure was not reached at the last repetition, the subjects performed additional reps until they could no longer perform an accepted repetition. Repetitions were performed with ~one sec concentric and two sec eccentric phases. Following the resistance exercise, all participants consumed a serving of whey protein (25 g protein, 3.6 g carbohydrate, 2.7 g fat (Bodylab, Hadsund, Denmark)) diluted in 400 mL of water. The protein supplement was given to standardize diet due to the otherwise long fasting period, but also to support the anabolic response to the resistance exercise bout.

Individual 10 RM was calculated using Brzyckis equation [37], based on a 5 RM test performed for each leg at the familiarization test.

### 2.7. Maximal Isometric Grip Strength

Maximal isometric grip strength was measured using a handheld dynamometer (PROcare ApS, Roskilde, Denmark). The subjects were seated holding the dynamometer in their dominant hand, while resting the other on their thigh. After a three second countdown, at the zero-mark, the subjects were instructed to squeeze the handle as hard as possible. All subjects completed three attempts with two minutes of rest between attempts. In case a subject improved peak torque during all three attempts, a fourth attempt was given. The highest torque was selected for further analysis.

### 2.8. Maximal Isometric Elbow Flexor Strength

Maximal isometric elbow flexor strength was measured in a custom made dynamometer. The subjects were seated on their knees with their dominant arm placed and strapped to the dynamometer with the elbow and the shoulder at a 90° angle. The subjects were instructed to keep their torso in a relaxed upright position and after a three second countdown, to pull as hard as possible against the dynamometer arm using their elbow flexor only. All subjects completed three attempts with two minutes of rest between attempts. In case a subject improved peak torque during all three attempts, a fourth attempt was given. The highest torque was selected for further analysis.

Rate of force development (RFD) was determined from the selected maximal voluntary contraction. This was done by taking the peak torque generated at the time intervals 0–50 and 0–100 ms relative to the onset of contraction.

### 2.9. Western Blotting

Approximately 20–50 mg of muscle tissue was separated from the initial muscle sample and homogenized in 1:20 mg/μL of lysis buffer containing 20 mM Tris, 50 mM NaCl, 50 mM NaF, 5 mM sodium pyrophosphate, 250 mM sucrose, 1% Triton-x100 and a cocktail of protease inhibitors (HALT). Following homogenization, the samples were centrifuged (13.000 rpm for 15 min) and the resulting supernatant was collected and aliquoted before storing at −80 °C. Protein concentrations were determined in triplicates using the Bradford method (Bio-Rad, Hercules, CA, USA).

The western blotting protocol followed standard procedures described in more detail in Oxfeldt et al. [38]. In short, protein samples were separated using a sodium dodecyl sulfate (SDS)-polyacrylamide gel and electroblotted onto polyvinylindene difluoride (PVDF) membranes before being blocked and incubated in primary antibodies overnight. Thereafter, membranes were washed, incubated in secondary antibodies and quantified with an UVP imaging system (UVP, Upland, CA, USA). All western blot data were normalized to an internal control followed by normalization to the total amount of protein for the respective sample determined by stain free technology. Data is presented as arbitrary values (RC at rest vs. PLA at rest and pre vs. post values). Western blotting outcome parameters were p-Akt Ser473, p-mTOR Ser2448, mTOR, p-p70S6K, p-P38 MAPK Thr180/182, p-FOXO3 Ser253, FOXO3a, FOXO1, MURF1, Atrogin-1, HSP27, HSP70, HSP90, Myogenin, MyoD and ER-β. The primary antibodies are specified in Table 2.

Importantly, using two different antibodies (ER-α antibody, cell signaling D6R2W #57761 and ER-α antibody, Santa Cruz (C-311) sc-787) we tried to determine ER-α, but the blots demonstrated either no binding or unspecific binding. For this reason, we did not include the ER-α data for further analysis.

### 2.10. Statistical Analyses

The statistical analysis was performed using a paired t-test and linear mixed model (STATA 15). Data is tested for normal distribution, and if absent, appropriate adjustments were carried out prior to the analysis, including log or square root transformation (p-mTOR, p-p706SK, MyoD, p-FOXO3, mTOR, p-Akt, p-p38MAPK, FOXO3a, HSP27, MURF1, FOXO1, HSP90, ER-β). All repeated measures (western blot targets, muscle strength and RFD) were analyzed with Treatment (RC, PLA) and Time (before and after the 14-day intervention) as fixed effects and subject ID as random effect. Data is presented as mean ± SD if not otherwise indicated. The statistically significant level was set at *p* < 0.05.

## 3. Results

### 3.1. Muscle Protein Content Following the RC/PLA Intervention Periods

Foxo1 (*p* = 0.033) and Foxo3a (*p* = 0.044) total protein content were significantly lower following 14 days of RC extract compared to PLA (Figure 3A,B). HSP27 (*p* = 0.023) total protein content was significantly higher following 14 days of RC extract compared to PLA (Figure 3C). Furthermore, a trend (*p* = 0.076) towards higher ER-β protein content was observed following RC extract compared to PLA (Figure 3D). No significant differences were observed for mTOR, MURF1, Atrogin-1, HSP90, HSP70, Myogenin or MyoD protein content.

### 3.2. Muscle Protein Expression in Response to Exercise Combined with Protein Supplementation

No significant treatment × time interaction was observed for muscle protein expression of the analyzed parameters in response to exercise. However, 3 h post exercise p-mTOR (RC *p* = 0.022, PLA *p* = 0.027), p-p70S6K (RC *p* < 0.001, PLA, *p* < 0.001) and HSP90 (RC *p* = 0.015, PLA *p* = 0.039) protein content were significantly higher compared to rest (Figure 4A,B,E,H). Furthermore, p-Akt was significantly higher 3 h post exercise following RC treatment only (*p* = 0.001), while Atrogin-1, HSP27, ER-β protein content were significantly lower 3 h post exercise following RC treatment only (*p* = 0.026, *p* = 0.003, *p* = 0.008) (Figure 4C,D,F–H). No significant difference was observed for any other proteins.

### 3.3. Changes in Muscle Strength and RFD Following the RC/PLA Intervention Periods

Muscle strength measured in the elbow flexor and hand grip strength were unchanged following 14 days of treatment with RC and PLA (Table 3). Similarly, we found no significant differences in RFD (Table 3).

## 4. Discussion

The present study investigated the effect of 14 days of fermented RC extract supplementation on signaling proteins related to muscle protein synthesis and breakdown at rest and in response to a resistance exercise bout. Here, we present novel data demonstrating that molecular markers of muscle protein degradation are downregulated following 14 days of RC extract compared to PLA.

With increasing age, muscle mass decreases, which over time may vastly affect an individual’s functional capacity and general health. In women, an accelerated loss of muscle mass and strength is observed during the transition into the postmenopausal state parallel with a marked decline in estrogen. The present findings highlight that providing early postmenopausal women with an isoflavones supplement (through RC extract with estrogenic activity), may reduce muscle protein breakdown and thereby be a possible treatment strategy to counteract loss of muscle mass in postmenopausal women.

### 4.1. Influence of RC Extract on Markers of Protein Degradation

We found the protein expression of FOXO1 and FOXO3a—two key transcription factors involved in protein degradation—to be significantly lowered following RC treatment compared to PLA. A large body of literature demonstrates that estrogen exerts anti-apoptotic actions [39], protects against skeletal muscle damage and promotes skeletal muscle regeneration [40]. A number of studies have investigated whether isoflavones from soy display the same protective effects as estrogen [23,24,25,41,42]. Ogawa et al. demonstrated that daidzein downregulated ubiquitin-specific protease 19 expression in murine C2C12 cells, and downregulated both ubiquitin-specific protease 19 mRNA and protein content in female mice following dietary daidzein consumption [25]. In line, feeding male rats a genistein rich diet for 24 days mitigated denervation-induced soleus muscle atrophy [24] and a 0.6% aglycone rich diet significantly attenuated denervation-induced decreases in muscle fiber atrophy in mice [23]. In contrast, two studies performed in postmenopausal women found no positive effect on muscle inflammation [41,42] or proteolysis [41] after 4 weeks of soy milk consumption. However, two major limitations exist in these human trials. First, the use of soy milk may not provide sufficient concentrations of bioavailable isoflavones. Secondly, both studies compared their soy milk treatment to a milk product, which may not be an adequate control when determining protein degradation, since milk protein stimulate muscle protein synthesis more than soy protein [43]. The latter may have masked the possibility for detecting positive effects of the isoflavones. Consequently, the present study is the first human study to demonstrate that intake of isoflavones through RC extract reduces markers of protein degradation in postmenopausal women.

Previous studies investigating effects of isoflavone intake on skeletal muscle used isoflavone supplements derived from soy. In the present study, we instead used a supplement containing isoflavones derived from fermented RC. The fermentation process is essential, as it removes the sugar residues and transforms inactive glycosides to active aglycones, which can then be absorbed by humans [33]. Since our fermented RC extract consists of 90% aglycones, the bioavailability of the RC isoflavones is much higher than in a traditional soy product. Therefore, the physiological effects are potentially much greater. The estrogenic effects of isoflavones are believed to be ER-β mediated, due to the strong binding affinity isoflavones have for this particular receptor [17]. In support of this, Ogawa et al. demonstrated that daidzein stimulated the transcriptional activity of ER-β in murine C2C12 cells and down-regulated ubiquitin-specific protease 19 expression [25]. In line with these findings, we found a trend towards a higher ER-β protein content following RC extract, which was observed concomitant with a significantly lower expression of FOXO1 and FOXO3a protein. To investigate whether a relationship existed between the expression of ER-β and the FOXO muscle protein degradation factors, we performed a Pearson correlation analysis, which demonstrated that ER-β expression at rest was negatively associated with both FOXO1 (R = −0.48, *p* = 0.03), and FOXO3A (R = −0.47, *p* = 0.03) protein expression. Furthermore, a Pearson correlation on the delta values (protein expression at rest following RC extract vs. PLA) showed a negative association between the change in protein expression of ER-β and the change in protein expression of FOXO1 (R = −0.75, *p* = 0.02), but not FOXO3A (R = −0.49, *p* = 0.14). Based on these data, we propose that the downregulation observed in the markers of protein degradation following RC extract is mediated through ER-β dependent signaling [22,44].

### 4.2. Influence of RC Extract on Heat Shock Proteins

HSPs are linked to the regulation of skeletal muscle remodeling. Most HSPs limit stress induced denaturation and aggregation of cellular proteins, and promote refolding and cellular homeostasis after stress [45]. With ageing, the heat shock response is significantly reduced and the HSP protein content is lowered [46], which may partly explain the impaired regeneration of skeletal muscle observed in older individuals [46]. Interestingly, a number of studies have demonstrated that estrogen influences HSPs positively [47,48]. For this reason, we investigated if RC extract similar to estrogen treatment would affect the expression of HSPs at rest and in response to exercise.

In the present study, HSP27 protein expression was significantly higher after 14 days of RC extract compared to PLA. HSP27 is involved in cell protection by directly stabilizing microfilaments and preventing protein aggregation [45]. Hence, an upregulation of HSP27 protein content may suggest that the skeletal muscle is less prone to protein degradation and aggregation following RC extract. To our knowledge, the present study is the first to investigate how ingestion of isoflavones influence HSPs in human skeletal muscle. Cell culture and animals studies investigating how estrogen influences HSPs have reported that estrogen treatment protects C2C12 cells against apoptosis through upregulation of HSP27 [48], and estrogen treated ovariectomized rats have elevated basal levels of HSP70 compared to placebo [47,49,50,51]. Thus, our novel data demonstrate that RC extract modulates specific HSPs in a way similar to estrogen treatment, which may have anti-apoptotic effects and protects postmenopausal women against muscle fiber atrophy.

### 4.3. Changes in Protein Expression in Response to Resistance Exercise Combined with Protein Supplementation

Resistance training is an effective strategy for increasing muscle mass and strength, and preventing the age-associated decline in skeletal muscle mass [5]. The response to resistance training is reduced in postmenopausal women [6,7,8], but treatment with HT seems to increase the response to resistance training [10]. The main way resistance training increases skeletal muscle mass is by increasing muscle protein synthesis through mTOR activation [52]. The phosphorylation of mTOR activates several downstream kinases, which increase translational efficiency and capacity [52]. Contrary to our hypothesis, we found no significant interaction of RC extract compared to PLA on signaling proteins related to muscle protein synthesis and breakdown in response to a single resistance exercise bout combined with protein supplementation. However, in line with previous investigations [53], p-mTOR and p-p70S6K protein content were significantly higher 3 h post exercise. Furthermore, p-Akt was significantly higher 3 h post exercise following RC extract only, while Atrogin-1, HSP27 and ER-β protein content were significantly lower following RC extract only. While these differences did not reach a significant interaction, possibly due to our small sample size, they may indicate that RC extract possibly influences signaling proteins involved in resistance training adaptations. Akt plays an important role in muscle hypertrophy [54]. In response to hormonal (estrogen) and growth factor stimulation, Akt activity is increased, which results in the activation of the mTOR-pathway and inhibition of the FOXO pathway [54]. This way, Akt both increases protein synthesis through mTOR phosphorylation and prevents FOXO from stimulating transcription of proteolytic ubiquitin ligases [54]. Thus, our data demonstrating a significant higher expression of p-Akt 3 h post exercise after RC extract, but not PLA, might suggest that RC extract enhances the anabolic response to resistance exercise combined with protein supplementation. In support of this, Atrogin-1 a protein located downstream of FOXO, was significantly lower 3 h post exercise after RC extract, but not PLA. Indeed this may be a result of p-Akt inhibiting FOXO specific transcription, resulting in lower atrogin-1 protein content. However, another potential explanation could be that the significant lower expression of FOXO3a observed at rest after RC extract compared to PLA regulates the expression of atrogin-1 in response to exercise, since FOXO3a interacts directly with atrogin-1 and thereby regulates its transcription [55].

While these molecular data look somewhat promising, they are only a snapshot of the accumulative response needed to increase muscle mass over time. Hence, a long term training study is needed to determine the influence of RC extract on skeletal muscle adaptations. Three long term training studies have investigated if supplementation with soy isoflavones enhanced adaptations to training, demonstrating no effect on lean body mass [28,29,30,31] or muscle strength [30]. However, as previously mentioned, while soy have high amounts of daidzein and genistein, the present fermented RC extract contains high amounts of the isoflavones formononetin and biochanin A as aglycones, which are highly bioavailable. Due to the differences in the chemical profile between RC and soy products a direct comparison between our results and previous studies cannot be made. Consequently, the influence of RC extract on adaptations to training is yet to be elucidated.

### 4.4. Changes in Muscle Strength and RFD

No significant effect of RC extract was found on muscle strength and RFD. These observations are in line with previous findings showing no beneficial effect of supplementing postmenopausal women with soy isoflavones in combination with exercise on muscle strength [30,31]. Some evidence suggests that estradiol positively influences muscle strength by affecting myosin binding and contractility directly [56]. In support, the force generation capacity in muscle fibers isolated from mice with estrogen deficiency are reduced [57], whereas estradiol treatment to ovariectomized mice prevents and restores loss of myosin-actin strong-binding and force generation [58]. Nevertheless, even though RC extract has estrogenic effects our results demonstrate that short term supplementation (14 days), does not enhance muscle strength or RFD. In contrast, a meta-analysis including 23 human studies has demonstrated that long term use (mean use of HT: 110 months) of HT has a small beneficial effect on muscle strength in postmenopausal women [9]. Hence, future studies are needed to determine whether long-term use of RC positively influence muscle strength in postmenopausal women.

### 4.5. Future Directions

A large body of evidence supports the use of isoflavones in the prevention of osteopenia [59] and to relive menopausal symptoms [60]. A risk assessment performed by the European Food Safety Authority (EFSA) concludes that intake of isoflavones up to 150 mg/day is safe for peri- and postmenopausal women [61]. These findings are supported by large epidemiological studies demonstrating that intake of isoflavones is not associated with estrogen related cancer [18,19,20], and may even reduce the risk of cardiovascular diseases [62].

We report, for the first time, that intake of isoflavones through a fermented RC extract reduces markers of protein degradation. This finding has important perspectives for many postmenopausal women, who suffer from a decline in muscle mass and muscle function during and after the menopausal transition.

A major strength of the present study is the design as a randomized placebo-controlled double-blinded cross-over trial. However, the short intervention period prevents us from drawing any conclusions on the long-term effect of RC extract on skeletal muscle. Future clinical trials need to enroll and randomize a larger number of subjects and prolong the intervention period to investigate effects of RC extract both with and without training on skeletal muscle, preferably using a double-blinded controlled design.

An interesting perspective that we have not addressed in the present article is the vascular benefits of isoflavones. Previous findings suggest that isoflavones improve endothelial function [15]. Endothelial function is not only important for overall health, but also an individual’s ability to respond to an exercise stimuli. In fact, recent studies have demonstrated that muscle fiber capillarization, which is substantially decreased during aging, may be predictive of the muscle hypertrophic response to resistance training in older adults [63,64]. If long-term intake of isoflavones improves endothelial function and muscle fiber capillarization, we speculate that this may indirectly have additive effects on the response to resistance training. Also, this may increase amino acid availability to the myofibers and thus stimulate muscle protein synthesis to a greater extent. For this reason, we encourage future studies to also evaluate endothelial function and muscle fiber capillarization.

## 5. Conclusions

This study demonstrates that 14 days of fermented red clover extract downregulates molecular markers of muscle protein degradation compared to placebo in early postmenopausal women.

## Figures and Tables

**Figure 1 nutrients-12-03587-f001:**
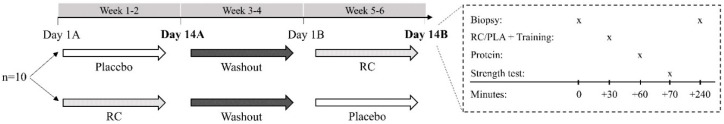
Overview of the study design with a detailed description of experimental day 14A and 14B. RC: red clover; PLA: placebo.

**Figure 2 nutrients-12-03587-f002:**
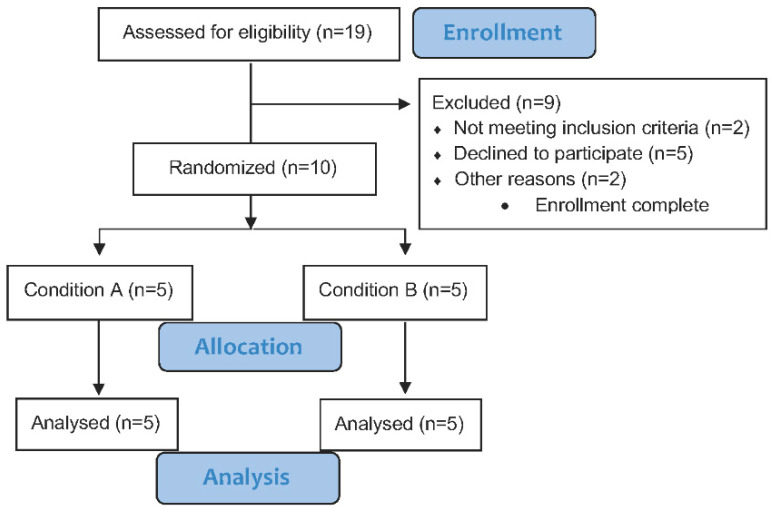
A detailed overview of the recruitment process.

**Figure 3 nutrients-12-03587-f003:**
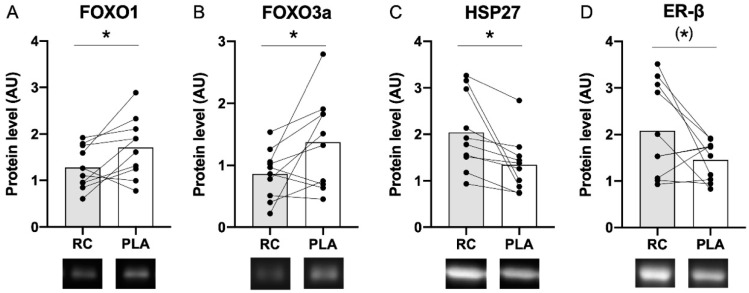
FOXO1 (**A**), FOXO3a (**B**), HSP27 (**C**), and ER-β (**D**) protein expression after 14 days of RC extract and PLA. The histogram represents mean values, while symbols and lines represent individual values. * Significant difference from PLA *p* < 0.05, (*) Tendency toward difference from PLA, *p* < 0.08.

**Figure 4 nutrients-12-03587-f004:**
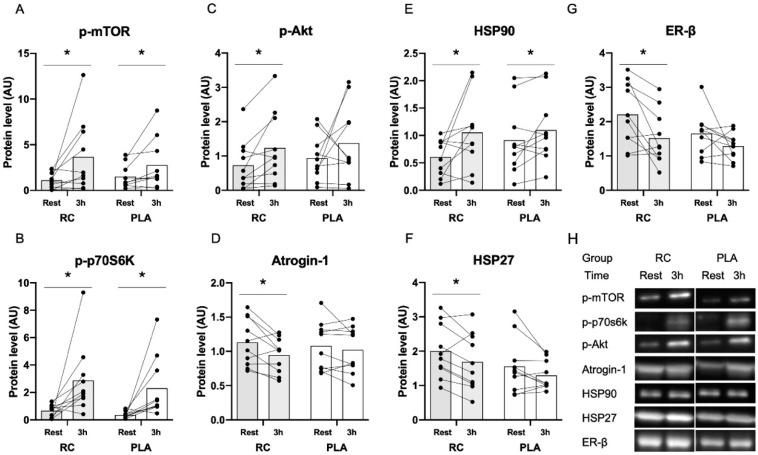
Phosphorylation of mTOR (**A**), phosphorylation of p70S6K (**B**), phosphorylation of Akt (**C**), Atrogin-1 (**D**), HSP90 (**E**), HSP27 (**F**) and ER-β (**G**) protein expression before and 3 h after resistance exercise. Representative western blots are shown in (**H**). The histogram represents mean values, while symbols and lines represent individual values. * Significant difference from rest *p* < 0.05.

**Table 1 nutrients-12-03587-t001:** Participant characteristics.

Subjects (*n* = 10)	
Age (y)	54 ± 4
Height (cm)	168 ± 6
Weight (kg)	70 ± 8
FM (kg)	26 ± 7
FFM (kg)	42 ± 4
Physical activity (min/week)	113 ± 55
Steps (Steps/day)	8550 ± 1892
Time since last bleeding (months)	18 ± 11

FM, Fat mass; FFM, Fat free mass. Data is presented as mean ± standard deviation.

**Table 2 nutrients-12-03587-t002:** Primary and secondary antibodies for western blotting.

Antibody	Physiological Process	Manufacturer	Catalog No.	Dilution	Blocking Agent	Secondary Antibodies	Dilution
P-Akt Ser^473^	Promotes cell survival by inhibiting apoptosis when phosphorylated	Cell Sig.	9271	1:1000	5% BSA, TBST	Anti-rabbit IgG, 7074S	1:2000
P-mTOR Ser^2448^	Promotes cell growth when phosphorylated	Cell Sig.	2971	1:1000	5% BSA, TBST	Anti-rabbit IgG, 7074S	1:5000
mTOR	Regulates cell growth	Cell Sig.	2972	1:1000	5% BSA, TBST	Anti-rabbit igG, 7074S	1:5000
P-p70S6K	Promotes cell growth, when phosphorylated	Cell Sig.	9234	1:500	5% BSA, TBST	Anti-rabbit IgG, 7074S	1:5000
P-P38 MAPK Thr^180/182^	A transducer of stress stimuli	Cell Sig.	9211	1:1000	5% BSA, TBST	Anti-rabbit IgG, 7074S	1:5000
P-FOXO3 Ser^253^	Promotes cell cycle arrest and apoptosis when dephosphorylated	Cell Sig.	9466	1:1000	5% BSA, TBST	Anti-rabbit IgG, 7074S	1:2500
FOXO3a	Promotes cell cycle arrest and apoptosis	Cell Sig.	2497	1:1000	5% BSA, TBST	Anti-rabbit IgG, 7074S	1:2000
FOXO1	Promotes cell cycle arrest and apoptosis	Cell Sig.	2880	1:1000	5% Milk, TBST	Anti-rabbit IgG, 7074S	1:2000
MURF1	Promotes muscle cell protein degradation	ECM Biosciences.	3401	1:1000	0.3% i-block, PBST	Anti-rabbit IgG, 7074S	1:2000
Atrogin-1	Promotes muscle cell protein degradation	Abcam	ab168372	1:1000	5% Milk, TBST	Anti-rabbit IgG, 7074S	1:2500
HSP27	Promotes cellular resistance	Abcam	ab109376	1:1500	5% Milk, TBST	Anti-rabbit IgG, 7074S	1:10,000
HSP70	Promotes cellular homeostasis	Abcam	ab181606	1:1000	5% Milk, TBST	Anti-rabbit IgG, 7074S	1:5000
HSP90	Promotes cellular homeostasis	Abcam	ab203126	1:1000	5% Milk, TBST	Anti-rabbit IgG, 7074S	1:5000
Myogenin	Promotes early myogenic differentiation	Milipore	3876	1:1000	5% Milk, TBST	Goat anti-mouse IgG	1:5000
MyoD	Promotes late myogenic differentiation	Abcam	ab126726	1:1000	5% BSA, TBST	Anti-rabbit IgG, 7074S	1:5000
ER-β	A receptor for estrogenic compounds	Abcam	ab3576	1:500	5% Milk, TBST	Anti-rabbit IgG, 7074S	1:5000

BSA, Bovine serum albumin; TBST, Tris buffered saline with tween. Cell Sig., Cell signaling Technology.

**Table 3 nutrients-12-03587-t003:** Changes in muscle strength and RFD.

	RC		PLA		
	Pre	Post	Pre	Post	Interaction
Grip strength (kg)	32.3 ± 7.6	32.2 ± 7.4	30.8 ± 6.3	32.4 ± 7.7	0.14
Elbow flexor strength (Nm)	39.0 ± 5.6	38.3 ± 7.5	37.4 ± 4.3	36.9 ± 4.6	0.93
RFD 0–50 ms (Nm)	5.6 ± 3.4	5.9 ± 4.4	7.0 ± 3.2	6.9 ± 3.7	0.80
RFD 0–100 ms (Nm)	11.1 ± 6.4	11.1 ± 6.0	12.7 ± 5.2	13.1 ± 5.1	0.83

Data is presented as mean ± standard deviation (SD). RFD: Rate of force development.

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
