# Peer review of "Influence of Fermented Red Clover Extract on Skeletal Muscle in Early Postmenopausal Women: A Double-Blinded Cross-Over Study"

_nutrients, 2020, doi:10.3390/nu12113587_

Round 1

Reviewer 1 Report

The authors present a crossover double blind study looking at the effects of two-weeks of fermented red clover extract consumption on markers of protein synthesis and degradation in early postmenopausal women. This study provides important initial insight into a potential treatment to address a critical need for treatment in women’s health. However, the short duration of the study hinders ability to determine whether any of the changes in protein expression lead to meaningful differences in skeletal muscle health or function. This study is a good step for future research that can more rigorously determine the effect of RC extract on skeletal muscle health and function.  However, there are several issues with the present study that the authors should address:

Major comments:

  • Figure 3 is incorrect – it is a duplicate of Figure 4.
  • Lines 147-148: The authors point to a rather wide range in daily isoflavone dosing. Is this because of differences in body weight? Is there a specific dose of isoflavone per bodyweight? Additionally, the authors do not report compliance. On the topic of dosage, were there any plasma measurements to indicate there was an increase in circulating isoflavone and/or relevant metabolites?
  • The authors report fat mass and fat free mass (Table 1) but do not report how this was measured.
  • The authors indicate there were no Group x Time interactions post-exercise, but what about effects of Group (i.e. Treatment) at rest? Did the authors compare protein expression between treatments at rest? The reviewer recommends the authors consider doing this as they make reference in the discussion to there being differences between treatment groups at rest.
  • Did the authors compare the relative change between resting and 3h post exercise for participants?
  • Lines 46 – 47 (under 4. Discussion) – the authors write data that “molecular markers of muscle protein degradation are positively regulated following 14 days of RC extract compared to PLA.” The reviewer assumes the authors are referring to decreases in Foxo1 and Foxo3a protein content as well as a decrease in Atrogin-1. To state that these markers are “positively” regulated would suggest that protein degradation is an inherently negative phenomenon. However, studies have demonstrated the importance of protein breakdown and protein turnover as a whole. Moreover, the authors do not demonstrate that an increase in Foxo1 and Foxo3a related degradation drives an aberrant decline in muscle mass through the transition of menopause. Thus, it is difficult to conclude that the changes in protein expression from RC consumption would “positively” affect protein quality.
  • Line 85 (of the discussion): The authors state “In line with these findings, we found a trend towards higher ER-B protein content following RC extract” but do not report this data in the data. The only comparisons the authors report are between Rest and Post Exercise.
  • How did the authors determine a 2 week washout period between interventions was an adequate amount of time to ensure there were no carry-over effects between the intervention?
  • Line 135 (in the discussion): The authors state that “a significant higher expression of p-Akt 3h post exercise after RC extract, but not PLA, may suggest that RC extract enhances the anabolic response to resistance exercise.” However, the authors also provided a protein drink immediately after exercise. Thus, the changes in protein expression were not exclusively elicited by resistance exercise and instead elicited by a combination of consuming a protein beverage and resistance exercise. Indeed, the addition of a protein beverage is rather confounding. Is it possible that RC consumption improved protein absorption? Additionally, the authors suggest at the end of the discussion (Lines 190-191) that RC may improve endothelial function and muscle fibre capillarization. One could reason that this would improve nutrient availability to the myofibres. Accordingly, the changes between rest and 3h post exercise (and beverage consumption) may be reflective of resistance exercise and instead protein consumption. The reviewer recommends the authors make relevant revisions to their discussion to reflect the confounding factor of the protein-rich beverage consumption.
  • It is important to note that, while the authors recognize that they only measure markers of protein turnover, there are no significant, positive functional (i.e. changes in protein synthesis or degradation, muscle quality/strength) outcomes. Thus, it is difficult to conclude whether or not any of these differences in protein expression between RC and placebo post exercise are beneficial or meaningful.

Minor comments:

  • Line 43: The use of the word “insignificant” is awkward in the following concentration – “possibly related to the insignificant circulating concentration of estrogen.” In the case, it might be clearer to use the word “low” instead of “insignificant.”
  • Line 25: The authors reference “Group x Time interaction.” This reviewer suggests for better clarity to use “treatment x time.”
  • Table 3 (Line 41) – why is “Grip Strength” bolded?
  • Line 59-60. The citation provided does not appear to support the notion that isoflavone intake of 20.3-178.7mg/day is protective. Table 2 of that citation does not demonstrate any protective effect of isoflavone intake. The CI for the hazards ratios (while below 1.0 in the highest quartile) all cross 1.0 indicative of no significant difference between that group and the reference group.
  • Table 2 – the authors report using antibodies such as P-P38 MAPK Thr or MURF. However, there are no data for that blot.
  • What is the concentration of the extract in the drink? Variability? Stability?
  • Were there any differences in activity levels within participants between treatment time?
  • Line 181 – there is a typo, “angel” should be “angle.”
  • Lines 94-95 of the discussion: the authors suggest that RC extract is mediated through ER-B dependent signaling, however, the authors do not provide specific mechanisms in which ER-B could be mediating this effect.

To conclude, while the authors’ present interesting data, there are several notable concerns that the authors should address in subsequent revisions.

Author Response

Dear Editor
The authors would like to thank the editor and reviewers for their relevant comments. It is our impression that addressing these comments have improved and clarified our work. We have addressed the reviewers’ comments in the following paragraphs. Minor issues such as grammar corrections have been integrated in the manuscript as recommended.

Mikkel Oxfeldt

Major comments:

  • Figure 3 is incorrect – it is a duplicate of Figure 4.

This confusing mistake is unfortunate. The mistake was apparently introduced by the system in the initial uploading process and in an automatic generation of a pdf-file of the manuscript. We have been unaware of this mistake, but agree that it is confusing and find it very unfortunate that important data is only described (paragraph 3.1-3.2), but not shown.

  • Lines 147-148: The authors point to a rather wide range in daily isoflavone dosing. Is this because of differences in body weight? Is there a specific dose of isoflavone per bodyweight? Additionally, the authors do not report compliance. On the topic of dosage, were there any plasma measurements to indicate there was an increase in circulating isoflavone and/or relevant metabolites?

All participants received the same amount of daily RC drink despite small differences in body weight. The RC drink is a natural product made from fresh red clover juice. The isoflavone content in red clover varies, which introduces differences in isoflavone content of each batch. The reported range is based on product analyses of the specific used batch. The company guarantee a minimum content.

Compliance was 100%. This important information have now been included in the Methods Section, page 4, line 151:

       Compliance was 100%.

We agree with the reviewer that measuring the circulating isoflavone metabolites would be of interest. We certainly will include this in any future study. We have collected the blood samples, but presently we have not been able to find a laboratory, which could analysis the blood samples for isoflavone metabolites.

  • The authors report fat mass and fat free mass (Table 1) but do not report how this was measured.

We apologize for this mistake. The participants went through a dual-energy X-ray absorption scan to determine their body composition. We have now included this in the method section, page 4, line 129-131:

Furthermore, they completed a physical activity questionnaire, a dual-energy X-ray absorption scan to determine body composition and a familiarization session to the laboratory tests.

  • The authors indicate there were no Group x Time interactions post-exercise, but what about effects of Group (i.e. Treatment) at rest? Did the authors compare protein expression between treatments at rest? The reviewer recommends the authors consider doing this as they make reference in the discussion to there being differences between treatment groups at rest.

There were no significant Group x Time interactions, but we did compare the protein expression levels at rest as the reviewer suggest (paragraph 3.1 and Figure 3, also see below), which we have referred to in the discussion. As mentioned above, the confusion about the results arise from the fact that Figure 3 somehow has been removed in the automatic generation of the manuscript pdf-file. 

  • Did the authors compare the relative change between resting and 3h post exercise for participants?

Figure 4 and paragraph 3.2 include the results/change from rest to post-exercise (i.e the relative change between rest and 3h post exercise)

  • Lines 46 – 47 (under 4. Discussion) – the authors write data that “molecular markers of muscle protein degradation are positively regulated following 14 days of RC extract compared to PLA.” The reviewer assumes the authors are referring to decreases in Foxo1 and Foxo3a protein content as well as a decrease in Atrogin-1. To state that these markers are “positively” regulated would suggest that protein degradation is an inherently negative phenomenon. However, studies have demonstrated the importance of protein breakdown and protein turnover as a whole. Moreover, the authors do not demonstrate that an increase in Foxo1 and Foxo3a related degradation drives an aberrant decline in muscle mass through the transition of menopause. Thus, it is difficult to conclude that the changes in protein expression from RC consumption would “positively” affect protein quality.

We have used the word ‘positively’ to indicate that the changes observed in foxo/atrogin-1 markers favors the RC drink over PLC when viewing our results in light of the changes in the expression of foxo/atrogin-1 markers and their association with muscle atrophy that are reported in the literature.

We do, however, agree with the reviewer that protein degradation must occurs as part of a health muscle homeostasis, and that we with this short term study are unable to correlate differences on protein expression levels with menopausal changes in muscle mass. Therefore, to accommodate you relevant comment we have throughout the manuscript revised the wording and excluded the wording “positively” in relation to downregulation of the signaling proteins coupled to muscle protein breakdown.

Abstract ln 25: “Conclusion: This study demonstrates that RC extract supplementation downregulates molecular markers of muscle protein degradation compared to placebo in postmenopausal women.”

Discussion ln 47: “molecular markers of muscle protein degradation are downregulated following 14 days of RC extract compared to PLA. “

Conclusion ln 195: “This study demonstrates that 14 days of red clover extract downregulates molecular markers of muscle protein degradation..”

  • Line 85 (of the discussion): The authors state “In line with these findings, we found a trend towards higher ER-B protein content following RC extract” but do not report this data in the data. The only comparisons the authors report are between Rest and Post Exercise.

Again, this confusion is related to the fact that Figure 3 has been replaced by Figure 4. The results depicted in the original Figure 3 is described in section 3.1 of the results and well as indicated in the figure legend of ‘wrong’ Figure 3.

  • How did the authors determine a 2 week washout period between interventions was an adequate amount of time to ensure there were no carry-over effects between the intervention?

The authors cannot be sure that the length of the washout period is long enough. Isoflavone metabolites are halved in the circulation after a variable time, but according to previous literature this occurs within hour (3-8h), which leaves very few metabolites after 14 days (1). Whether the impact of the isoflavones on the muscle tissue, the receptor and protein expression levels, etc. will remain after the 14 days is presently unknown. We found no difference in protein expression between the participants ingesting the RC drink in the first intervention period compared to the participants ingesting the RC drink in the second intervention period indicating that the wash out period was of an appropriate length.

  • Line 135 (in the discussion): The authors state that “a significant higher expression of p-Akt 3h post exercise after RC extract, but not PLA, may suggest that RC extract enhances the anabolic response to resistance exercise.” However, the authors also provided a protein drink immediately after exercise. Thus, the changes in protein expression were not exclusively elicited by resistance exercise and instead elicited by a combination of consuming a protein beverage and resistance exercise. Indeed, the addition of a protein beverage is rather confounding. Is it possible that RC consumption improved protein absorption? Additionally, the authors suggest at the end of the discussion (Lines 190-191) that RC may improve endothelial function and muscle fibre capillarization. One could reason that this would improve nutrient availability to the myofibres. Accordingly, the changes between rest and 3h post exercise (and beverage consumption) may be reflective of resistance exercise and instead protein consumption. The reviewer recommends the authors make relevant revisions to their discussion to reflect the confounding factor of the protein-rich beverage consumption.

This is a very relevant suggestion. We have now made this more clear in the discussion and result section by adding: “combined with protein supplementation” when the resistance exercise is mentioned.

We have also added a sentence on the considerations the reviewer had on the possible role RC may have on nutrient availability:

If long-term intake of isoflavones improve endothelial function and muscle fibre capillarization, we speculate that this may indirectly have additive effects on the response to resistance training. Also, this may increase amino acid availability to the myofibers and thus stimulate muscle protein synthesis to a greater extend. For this reason, we encourage future studies to evaluate endothelial function and muscle fiber capillarization.

  • It is important to note that, while the authors recognize that they only measure markers of protein turnover, there are no significant, positive functional (i.e. changes in protein synthesis or degradation, muscle quality/strength) outcomes. Thus, it is difficult to conclude whether or not any of these differences in protein expression between RC and placebo post exercise are beneficial or meaningful.

We agree with the reviewer in this matter, and we are fully aware that any change in muscle mass or functionality would likely not occur within a 14-day time period. Thus, we have throughout the manuscript underlined that this is a pilot study generating ideas for future long-term RCT studies. 

Minor comments:

  • Line 43: The use of the word “insignificant” is awkward in the following concentration – “possibly related to the insignificant circulating concentration of estrogen.” In the case, it might be clearer to use the word “low” instead of “insignificant.”

We have changed to word to ‘low’ as suggested by the reviewer.

  • Line 25: The authors reference “Group x Time interaction.” This reviewer suggests for better clarity to use “treatment x time.”

We have changed the wording to treatment x time throughout the manuscript.

  • Table 3 (Line 41) – why is “Grip Strength” bolded?

Grip strength is not supposed to be bolded, but this again is an edit that occurred automatically by the system as the manuscript was initially submitted. The same problem appears in Table 1 with the word ‘ Age’. We ensure that this will be corrected in the proof-reading process.

  • Line 59-60. The citation provided does not appear to support the notion that isoflavone intake of 20.3-178.7mg/day is protective. Table 2 of that citation does not demonstrate any protective effect of isoflavone intake. The CI for the hazards ratios (while below 1.0 in the highest quartile) all cross 1.0 indicative of no significant difference between that group and the reference group.

This is a great observation made by the reviewer! We have interpreted the results one-step too far. We have changed the sentence accordingly. Line: 57 and forward.

  • Table 2 – the authors report using antibodies such as P-P38 MAPK Thr or MURF. However, there are no data for that blot.

We understand the confusion. These proteins showed no significant difference. Therefore, we did not include them in the figures. To make this more transparent to the reader, we have added a statement in both paragraph 3.1 and 3.2:

No significant difference was observed for any other proteins.”

  • What is the concentration of the extract in the drink? Variability? Stability?

We have not had the isoflavones content of this exact RC drink analysed, but a number of previous batches have been analysed to guarantee the minimum content of isoflavones. Yet, all RC drinks in this study came from the same batch ensuring an identical isoflavone content and eliminating viability between participants. The product is packed in air-tight sackets (bag-in-box), which insures a stable product and extends shelf-life.

  • Were there any differences in activity levels within participants between treatment time?

We did not observed any difference in accumulated steps per day for the participants between treatment one or two. The total number of steps are noted in Table 1.  

  • Line 181 – there is a typo, “angel” should be “angle.”

Thanks for noticing this. We have corrected accordingly.

  • Lines 94-95 of the discussion: the authors suggest that RC extract is mediated through ER-B dependent signaling, however, the authors do not provide specific mechanisms in which ER-B could be mediating this effect.

This is an interesting comment. We have now added a citation on two interesting studies looking into this mechanism.

To conclude, while the authors’ present interesting data, there are several notable concerns that the authors should address in subsequent revisions.

We hope that you are satisfied with the revised version of the manuscript

References:

  1. Shelnutt SR, Cimino CO, Wiggins PA, Ronis MJ, Badger TM. Pharmacokinetics of the glucuronide and sulfate conjugates of genistein and daidzein in men and women after consumption of a soy beverage. Am J Clin Nutr. 2002;76(3):588-94.

Reviewer 2 Report

The manuscript is very interesting, it is a good pilot for further research - it requires an increase in the research group and a longer observation time in order to translate the conclusions to the entire population.

The presented manuscript takes up an important topic - the search for medicinal substances that improve the function of skeletal muscles in early menopause. The results indicate that the tested extract may be an alternative to hormone therapy.
The study was designed as a randomized, crossover study which adds value to the work. Only a small group size requires continued research. The results and conclusions are in line with the goals set. The text is clear and easy to read.

Author Response

Thank you so much!

We are extremely glad for your interpretation of the paper.

Really interesting pilot data. Hopefully, we will be able to perform a larger clinical trial in a near future.

Thank you for reviewing our paper.

All the best

Mikkel

Round 2

Reviewer 1 Report

The reviewer appreciates the work into revising the manuscript. The reviewer agrees the manuscript has improved. 

The reviewers would encourage the authors to consider publishing graphs of the Western blot data that were not significantly different. For example, given the fact that MURF is downstream of FOXO. Given that RC changed FOXO levels but did not change MURF levels, is interesting and can provide readers more insight into the potential mechanisms in which RC may influence protein turnover. 

Both MURF1 and atorgin-1 both rely upon FOXO3a to initiate their transcription. These disparate results are interesting and could be discussed.

Overall, the reviewer does not have any more comments. The reviewer is concerned about bioavailability, consistency of the dose, and the absence of a significant functional outcome. However, this is a pilot study; hopefully these data will provide necessary evidence to support a grant to study RC in a more comprehensive RCT.

Author Response

Dear reviewer

Once again, thank you for your constructive comments!

We have now read through our paper and corrected some grammar and spelling issues. Furthermore, to address your comment on our non-significant findings. We have made a clear statement in our results section, mentioning the specific proteins of concern: 

"No significant differences were observed for mTOR, MURF1, Atrogin-1, HSP90, HSP70, Myogenin or MyoD protein content"

Finally, we have added a statement on why we chose to give the participants a protein supplement: 

"The protein supplement was given to standardize diet due to the otherwise long fasting period, but also to support the anabolic response to the resistance exercise bout"

Thank you for reviewing our paper.